# The Reflux and BariClip: Initial Results and Mechanism of Action

**DOI:** 10.3390/jcm11226698

**Published:** 2022-11-12

**Authors:** Patrick Noel, Laurent Layani, Thierry Manos, Mourad Adala, Sergio Carandina, Anamaria Nedelcu, Marius Nedelcu

**Affiliations:** 1Emirates Specialty Hospital, Dubai 505240, United Arab Emirates; 2Mediclinic Airport Road Hospital, Abu Dhabi 48481, United Arab Emirates; 3ELSAN, Clinique Bouchard, 13000 Marseille, France; 4Al Sharq Hospital, Fujairah 00000, United Arab Emirates; 5Clinique El Manar, Tunis 2092, Tunisia; 6ELSAN, Clinique Saint Michel, Centre Chirurgical de l’Óbesite, 83100 Toulon, France

**Keywords:** bariclip, laparoscopic bariclip gastroplasty, reflux, gerd, laparoscopic sleeve gastrectomy

## Abstract

Introduction: Laparoscopic BariClip Gastroplasty (LBCG) represents a new bariatric procedure that mimics the principle of the Laparoscopic Sleeve Gastrectomy (LSG), but using a completely reversible mechanism, which is essential for gastroesophageal reflux disease (GERD). The purpose of our study was to evaluate the evolution of GERD following the initial experience with LBCG. Methods: The first 43 obese patients who underwent LBCG performed by the same surgeon in two different medical centers in May 2018–December 2019 were included in the current study. Twelve patients had issues of reflux, regularly receiving PPIs (proton pump inhibitors) treatment in eight cases, and occasionally in four cases. Thirty-two patients completed the follow-up at one year and the GERD was evaluated using the PPI medications and the GerdQ. Results: The median preoperative GerdQ score was (14.58 ± 1.9). Three patients out of the twelve who had complained about preoperative GERD did not consent to the one year follow-up form. For the rest of nine patients, the median post-operative GerdQ score was (10.11 ± 3.2). The PPIs were used at one year follow-up in six patients: four with occasional use, one patient with regular use showing no improvement, and one who experienced de novo GERD symptomatology (3.1%). No statistically significant difference between the groups was recorded in terms of GERD. We recorded no intraoperative complications. No case of erosion occurred in the post-operative period, but we encountered two cases of slippage. One additional BariClip was removed at 14 months. Conclusion: LBCG represents a new bariatric procedure that mimics the principle of the laparoscopic sleeve gastrectomy, but with a completely reversible mechanism. Even with limited cases, our experience reports several mechanisms of action that will be evaluated and discussed in further prospective clinical trials. After this preliminary clinical study, LBCG’s effects on GERD and its safety are highly encouraging.

## 1. Introduction

Nowadays, sustainable weight loss and reduction in comorbidities in morbidly obese patients can be achieved using bariatric surgery. Among its tools, the most-performed bariatric procedure worldwide, during the past decade, was the laparoscopic sleeve gastrectomy (LSG) [1].

The development of LSG to the detriment of the laparoscopic Roux-en-Y gastric bypass is explained by its absence of dumping syndrome, malabsorption, marginal ulcers, small bowel obstruction, and internal hernia, which are the feared side effects of bypass procedures. At the same time, LSG simply allows a better quality of life (QoL), thus being preferred over gastric banding [2,3].

Looking at the complications following LSG, the main risk is increasing GERD (gastro-esophageal reflux disease) and PPI (proton pump inhibitors)-dependency or developing de novo GERD [4], causing the development of Barrett’s esophagus (BE) in up to 18% of patients [5]. The mechanisms which the LSG impacts on GERD are contradictory [4,5,6,7,8]. This surgery causes an alteration of the angle of His, hypotony of the lower esophageal sphincter (LES) due to the division of the muscular sling fibers, reduction in the gastric volume and consequently elevated intragastric pressure, and lower ghrelin level conducing to dysmotility. In these cases, the quality of life can be deteriorated, forcing patients into long-term medical therapy, or conversion to Roux-en-Y gastric bypass (RYGB) [9].

Accordingly, over the past decade, an important interest has grown in stomach-sparing procedures and several techniques have emerged with an important focus on bariatric endoscopy using different techniques of gastroplasty to treat morbid obesity. Even though the initial results were encouraging [10], their durability and the long-term results are debatable [11]. Laparoscopic BariClip gastroplasty (LBCG) could answer the need for a no-resection procedure. LBCG offers a higher restriction than the endoscopic procedures with the advantage of the reversibility by clipping without cutting the stomach; the LBCG procedure almost replicates the effectiveness of the LSG with minimal complications [12,13]. The procedure is performed using a nonadjustable clip that is vertically placed parallel to the lesser curvature. The purpose of our study was to evaluate the development of GERD following the initial trial with BariClip.

## 2. Materials and Methods

Forty-three patients who underwent the Laparoscopic BariClip Gastroplasty study trial between May 2018 and December 2019 were included in the current study. All the procedures were performed by the same surgeon in two different hospitals. Informed consent was obtained from all individual participants for this study.

GERD was evaluated both in the preoperative and post-operative period based on the presence of a PPI treatment and the GerdQ questionnaire [14]. The PPI use was defined as regular when the patient confirmed almost the daily use (min four times a week), or occasional when the use was less or equal to three times a week. A systematic preoperative upper endoscopy was performed and none of the patients presented a hiatal hernia measuring more than 2 cm on the preoperative study. In four cases, a small sliding hiatal hernia was described on the endoscopy report but not confirmed by the upper GI swallow (in the Trendelenburg position), nor by the intraoperative findings.

Our primary objective of the study was to evaluate the GERD after LBCG. The secondary objective was to report the findings from the early stage (within 30 days) and intermediate complication rates following the new bariatric procedure. All procedures performed in this clinical trial that involved human participants were conducted in accordance with the ethical standards of the institutional and/or national research committee and with the 1964 Helsinki Declaration and its later amendments and comparable ethical standards.

### Surgical Technique

The LBCG surgical technique has already been published in previous studies [15]. We start by creating a small opening at the angle of His using an articulated dissector, developing into an opening of 3 to 4 cm on the greater curvature, just underneath the incisura angularis. We then pass the articulated dissector at 90° into the lesser sac, to the left of the gastric vessels, and come out at the angle of His. The size of the pouch is calibrated using a 36 F bougie. The introduction of the BariClip into the peritoneal cavity is conducted through a 12 mm trocar. The locking of the BariClip around the stomach parallelly to the lesser curvature excludes a large lateral segment of the stomach, maintaining only a small medial pouch. The BariClip is secured to the gastric wall both anteriorly and posteriorly with stiches placed at various levels of the stomach. We recommend suturing using the left indentations in order to preserve the vessels of the lesser curvature situated next to the right indentations.

The patients are encouraged to start early ambulation and liquid intake is allowed 6 h after the procedure. They are released from the hospital the next day with a prescription of PPI (Pantoprazole 40 mg) for 30 days and Clexane 0.4 im for 2 weeks. The nutritional protocol, common to most bariatric procedures, begins with 2 weeks of liquid feeds, proceeding with 2 weeks of soft diet before reintroducing solid food. 

## 3. Results

The preoperative GERD signs were present in 12 patients out of the 43 undergoing a LBCG procedure, with regular PPI treatment in 8 patients and occasional use in 4 patients. The median preoperative GerdQ score was (14.58 ± 1.9). The preoperative endoscopy revealed grade A esophagitis for seven patients and grade B esophagitis for four patients.

Thirty-two patients out of the forty-three patients who underwent LBCG completed the follow-up at one year. Three patients out of the twelve who had complained about preoperative GERD did not complete the one year follow-up form. For the other nine patients, the median post-operative GerdQ score was (10.11 ± 3.2). The progression of the score for each patient is illustrated in Figure 1.

Regarding the use of PPI medication at one year follow-up, the acid reflux was present in six of the patients who were taking PPI occasionally (Figure 2). Four out of six of these patients reported effectiveness of PPI in preoperative phase. One patient showed no improvement in GERD symptomatology, while the remaining one patient did not suffer from of any symptoms during preoperative phase, but the patient experienced de novo GERD symptomatology (3.12%). The results concerning the evolution of GERD and PPI treatment following BariClip were summarized in Table 1. No statistically significant difference between the groups was recorded in terms of GERD.

### Post-Operative Complications

We recorded no intraoperative complications. In the early phase of the post-operative period, one case of slippage (2.3%) occurred. For a case with major slippage, a laparoscopic exploration was needed and the BariClip was removed. There was no case of bleeding or leakage in the early post-operative period (30 days).

Between 6 and 12 months after the procedures, seven patient (21.8%) complained about minor discomfort or moderate pain aggravated in different positions related to the BariClip incisions. In one case, a minor slippage was detected, and the medical treatment given to the patient was successful to overturn nausea and pain. As part of follow-up with the same patient, an upper endoscopic control was performed at 7 months after surgery, identifying the diverticulum with no other pathological concerns. At 14 months post-operatively, the patient reached a BMI of 22 and the decision was made to remove the BariClip. For all the other patients, the pain was relieved by different medication, except in one patient with excessive weight loss (post-op BMI-21) and we decided to conduct a BariClip removal after 14 months. Out of 32 patients who signed up one year follow-up, only three BariClip (9.4%) were removed successfully with no additional complications.

## 4. Discussion

In the past decade, the increasing number of bariatric procedures indicates that LSG is the most frequently performed by surgeons globally. This progression is supported by two factors: the decrease in the number of complications (especially in long-term comparing with LRYGB), and the better quality of life reported by patients (compared to the gastric banding). The Achille’s tendon of LSG remains the GERD, which, for some patients, represents both a long-term complication and a major factor in decreasing the quality of life. For surgeons, it generates a continuous debate regarding the risk of Barrett’s esophagus and its evolution.

Although scarce, emerging data can be found in several published studies concerning the development of Barrett’s esophagus after sleeve gastrectomy, in patients with no preoperative history of the disease. The diagnosis is established on mid-term follow-up upper GI endoscopies. In the study of Genco et al. [5], 19 out of 110 patients (17.2%) presented Barrett’s esophagus on biopsy at a mean of 58 months post-LSG. Among these 19 patients, only 14 (73.6%) had GERD symptoms, showing no correlation between the presence of symptoms and the severity of the esophageal disease. Similarly, Sebastianelli et al. [15] encountered in their multicenter study a prevalence of 18.8% for Barrett’s esophagus, in patients with normal esophagus having proceeded with LSG. In 35% of these cases, PPI treatment was prescribed, while only one patient complained of GERD symptoms. The difference among centers was non-significant and there was no dysplasia noted. Secondarily, they reported another correlation between the presence of BE and weight loss failure (defined as EWL < 50%).

Some modifications to the standard LSG have been proposed in the literature to control post-operative GERD. Nocca et al. [16] described the Nissen-Sleeve (N-Sleeve), a modification to the usual surgical technique of LSG by adding a Nissen fundoplication to minimize both leak risk and GERD. The Nissen-Sleeve was criticized for not achieving similar results in terms of weight loss due to the non-resected fundus used for the valve.

Nowadays, the number of patients hesitating or refusing the choice of LSG because of long-term GERD cannot be neglected. There are still many patients that are considering LSG a mutilating procedure by its resection mechanism. To overcome these limitations, Jacobs et al. [10] endorsed the Laparoscopic BariClip Gastroplasty as the alternative. In the previous study [13], we reported that LBCG offers an acceptable quality of life with the same principles as LSG and presents several advantages. There is no risk of leak or bleeding, and the LBCG remains a reversible bariatric procedure, and in the case of invalidating GERD, the BariClip can be removed with no complications—allowing, as needed, an antireflux procedure to be performed on a normal stomach.

Regarding the evolution of GERD following LBCG, there is limited data in medical literature, however, but several mechanisms of action can be discussed.

During the dissection, there is no anatomical damage of the ring and longitudinal fibers at the gastroesophageal junction. During the preclinical studies, a particular attention was paid to the closing pressure of the device. The BariClip is designed to minimize the closing force so that the limbs will simply oppose the anterior and posterior walls of the stomach to minimize the possibility of erosions and ischemia; the closing of the BariClip has been designed to be a low-pressure system.Minimal dissection of the His angle with no damage of the phreno-esophageal membrane. The posterior passage is realized with a minimal dissection in an avascular plane.The presence of a distal opening at the bottom of the BariClip makes the procedure a low-pressure system. It balances the pressures with the excluded part of the stomach, thus allowing for a possible acid or bile reflux to go towards the fundus, to the excluded portion of the stomach and not the esophagus, like in the illustrated upper GI swallow (Figure 2).A careful selection of patients is important. LBCG has not been performed in patients with large hiatal hernia.Compared with LSG, during LBCG, there is no section of the longitudinal fibers of the stomach. This type of gastric muscle fibers is involved in the gastric emptying. The shape of the gastric tube and the asymmetry between the anterior and the posterior parts could be an important factor for GERD. This risk factor is completely absent for LBCG.

Considering all these elements, our current experience reports a resolution of GERD in four out of nine patients, an improvement in four patients, one patient had no modification of his GERD symptomatology, and one patient developed de novo GERD. No statistically significant difference between the groups was recorded in terms of GERD. We are also convinced that the surgical technique plays a major role in the occurrence of reflux following different bariatric surgeries. For this reason, we decided to include in the current study the patients operated by a single surgeon.

One of the major limitations of our study, as for many other studies, is represented by the GERD evaluation. Measuring GERD is a difficult challenge. Chan et al. [17] showed the difficulty between self-reported reflux symptoms and their correlation with objectified reflux in 336 patients. Only half of the patients who claimed to have GERD were confirmed positively by tests such as the 24 h pH-metry. In our study, all the procedures were performed in a private practice, all the costs being supported by the patients with no possibility to perform an additional pH-metry study. Another limitation of the current study is represented by the reduced number of patients included, as for every new innovative bariatric procedure. Thus, we decided to use only descriptive analytics for the 12 patients with reflux symptoms and any statistical correlation was considered futile.

## 5. Conclusions

LBCG represents a new bariatric procedure that mirrors the principle of the laparoscopic sleeve gastrectomy, but with a complete reversible mechanism. Despite the limited number of cases, our experience reports several mechanisms of action that will be evaluated and discussed in further prospective clinical trials. After this preliminary clinical trial, the LBCG’s effects on GERD and its safety are highly encouraging. The procedure remains experimental and is under evaluation in a few clinical trials worldwide.

## Figures and Tables

**Figure 1 jcm-11-06698-f001:**
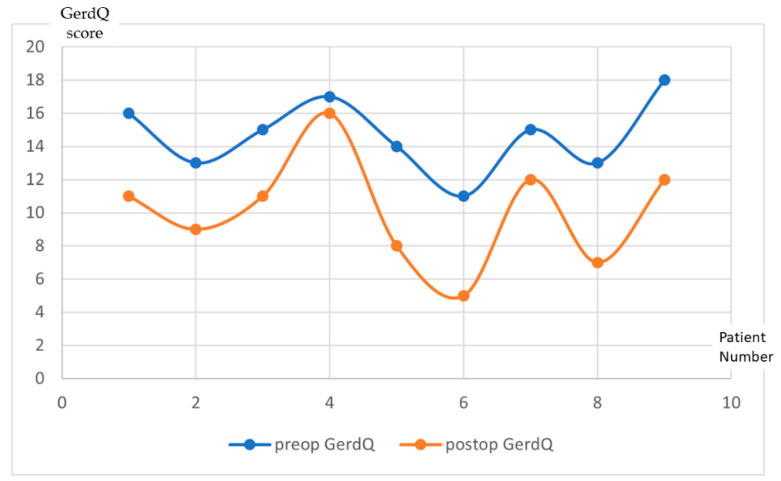
GerdQ results evolution at one year follow-up after LBCG. (x—Patient number; y—GerdQ score).

**Figure 2 jcm-11-06698-f002:**
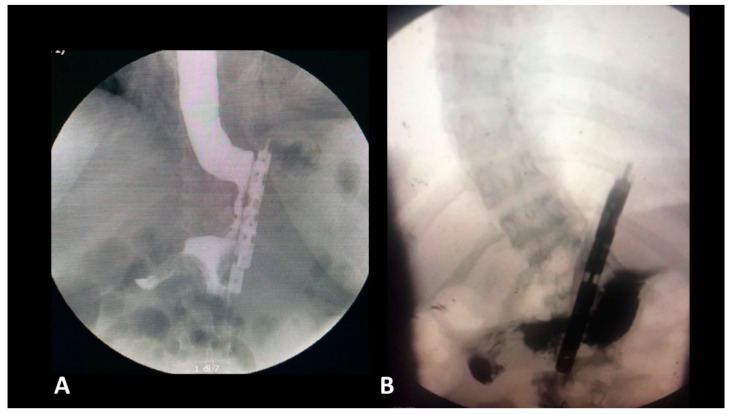
Postoperative gastrografin swallow (**A**) no reflux; (**B**) with reflux in the gastric remnant.

**Table 1 jcm-11-06698-t001:** GERD evolution and PPI treatment.

	Preoperatively	One Year Postoperatively
GERD with systematic PPI use	8 patients	1 patient
GERD with occasional PPI use	4 patients	6 patients
De novo GERD	-	1 patient

PPI, Proton Pump Inhibitors; GERD, Gastro Esophageal Reflux Disease.

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
