# Peer review of "The Reflux and BariClip: Initial Results and Mechanism of Action"

_jcm, 2022, doi:10.3390/jcm11226698_

Round 1
Reviewer 1 Report
First of all, thank you for giving an opportunity to review this interesting paper. Below are my comments.
Major points:
# (page 2, line 64, 66 and 81)
The work seemed to be conducted as part of a clinical trial. The authors need to describe the registry details in the M and M section.
# As the authors pointed out in the discussion part, the evaluation of GERD is challenging. In this study, the primary measure is to evaluate the impact of LBCG on GERD. The authors performed a systematic (meaning in all 43 patients?) pre-OP endoscopy (which is VERY GOOD). Then, it would be much more informative if the authors add the information about the degree of esophageal mucosal injury (LA classification, for example) and include those who had either of the following conditions for the analysis; (1) regular PPI use, (2) GERD-like symptoms (assessed by the GERD-Q questionnaires) and (3) esophageal mucosal injury assessed by endoscopy.
# In comparison with LSG, the readers (including myself) would have an interest about the impact of LBCG on de novo GERD in those who were NOT associated with GERD pre-OP. If the authors will also be able to include the related data, the study will be more valuable.
Minor points:
# (page 2, line 53)
Roux Y Gastric bypass (RYGB)→Roux-en-Y gastric bypass (RYGB)
# (page 2, line 94)
Since one of the purposes of this trial is to validate the complication rate after the technique, the slippage rate (“approximately 3%”) should not be written here.
# (page 3, line 105)
IPP→PPI
# (page 5, line 182)
Hiss→His
Author Response
First of all, thank you for giving an opportunity to review this interesting paper. Below are my comments.
We thank the reviewers for their fair and very constructive feedback. We have done the appropriate modifications according to our experience and convictions. We are convinced that by the modifications done to the manuscript according to your suggestions we have highly improved the quality of our paper.
Major points:
# (page 2, line 64, 66 and 81)
The work seemed to be conducted as part of a clinical trial. The authors need to describe the registry details in the M and M section.
Unfortunately, this was not conducted as a clinical trial as the study was conducted in 2 different hospitals from two different countries where the procedure is always considered as investigational procedure.
# As the authors pointed out in the discussion part, the evaluation of GERD is challenging. In this study, the primary measure is to evaluate the impact of LBCG on GERD. The authors performed a systematic (meaning in all 43 patients?) pre-OP endoscopy (which is VERY GOOD). Then, it would be much more informative if the authors add the information about the degree of esophageal mucosal injury (LA classification, for example) and include those who had either of the following conditions for the analysis; (1) regular PPI use, (2) GERD-like symptoms (assessed by the GERD-Q questionnaires) and (3) esophageal mucosal injury assessed by endoscopy.
The preoperative endoscopy is mandatory for our preoperative work-up both for sleeve and bypass for all patients. We completely agree with your suggestion but we need more additional data regarding the endoscopic finding after one year, and unfortunately very limited number of patients (4) have had a control endoscopy after one year.
# In comparison with LSG, the readers (including myself) would have an interest about the impact of LBCG on de novo GERD in those who were NOT associated with GERD pre-OP. If the authors will also be able to include the related data, the study will be more valuable.
Please find the following information about the evolution of GERD and the novo GERD in the lines 122- 127 (page 3): “Regarding the use of PPI medication at one year follow-up, the acid reflux was present in 6 of the patients who were taking PPI occasionally. Four out of 6 of these patients reported effectiveness of PPI in preoperative phase. 1 patient showed no improvement of the GERD symptomatology while the remaining 1 patient did not suffer from of any symptoms during preoperative phase, but the patient experienced de novo GERD symptomatology (3.12 %).”
Minor points:
# (page 2, line 53)
Roux Y Gastric bypass (RYGB)→Roux-en-Y gastric bypass (RYGB)
Thank you very much for your suggestion, the text was modified accordingly.
# (page 2, line 94)
Since one of the purposes of this trial is to validate the complication rate after the technique, the slippage rate (“approximately 3%”) should not be written here.
Thank you very much for your comment. This information was removed from this part of the manuscript and included in the results’ section.
# (page 3, line 105)
IPP→PPI
Thank you for the input, we apologies for this error
# (page 5, line 182)
Hiss→His
The name was modified according to your suggestion.
Reviewer 2 Report
This study by Noel et al investigates the effects of Laparoscopic BariClip Gastroplasty on Gastroesophageal reflux disease (GERD). Gerd development has been previously associated with the common bariatric surgery procedure called Laparoscopic Sleeve Gastrectomy. Although this is a preliminary study in a small cohort of patients and with short-term follow-up, it illustrates that BariClip may be an alternative procedure that will minimize the risk of GERD development or improve pre-operative GERD.
The following issues should be addressed before publication.
Major points
- Statistical analysis of the results is not present in the manuscript. For example, is the difference in the GerdQ score pre- and post-op statistically significant? In this line, figure 1 should be accompanied by a graph-box plot depicting median scores. Moreover, you can explain briefly what is the GerdQ score and how it is calculated.
- The presentation of the results regarding PPI medication is confusing. Figure 2 should be accompanied by a table summarizing the PPI medication pre- and post-op (frequent, occasional user and de novo cases after BariClip.
- BMI data are not present in the manuscript. Is GERD linked to BMI in this cohort? Is there an association between weight loss (reduced BMI) and GERD remission?
Minor points
1. Please define GERD abbreviation in the title
2. lines 14-15 (abstract) please clarify the meaning
3. line 45 gastric banding or gastric bypass?
4. The introduction should include more references and give a better presentation of Gerd associated with bariatric surgery.
5. The list of complication can be presented in a table.
6. Reference should be corrected (for example refs 7 and 8)
7. The abstract should be edited for clarity.
Author Response
This study by Noel et al investigates the effects of Laparoscopic BariClip Gastroplasty on Gastroesophageal reflux disease (GERD). Gerd development has been previously associated with the common bariatric surgery procedure called Laparoscopic Sleeve Gastrectomy. Although this is a preliminary study in a small cohort of patients and with short-term follow-up, it illustrates that BariClip may be an alternative procedure that will minimize the risk of GERD development or improve pre-operative GERD.
We thank the reviewers for their fair and very constructive feedback. We have done the appropriate modifications according to our experience and convictions. We are convinced that by the modifications done to the manuscript according to your suggestions we have highly improved the quality of our paper.
The following issues should be addressed before publication.
Major points
- Statistical analysis of the results is not present in the manuscript. For example, is the difference in the GerdQ score pre- and post-op statistically significant? In this line, figure 1 should be accompanied by a graph-box plot depicting median scores. Moreover, you can explain briefly what is the GerdQ score and how it is calculated.
Thank you very much for your comment. We have discussed this issue with our statistics department and considering the limit number of patients included in this study (43) and especially those with reflux symptoms (only 12 patients) any statistical correlation was considered futile. Their recommendation was to use descriptive analytics. For more information about the evolution of GERD and the novo GERD please find the following paragraph in the lines 122- 127 (page 3): “Regarding the use of PPI medication at one year follow-up, the acid reflux was present in 6 of the patients who were taking PPI occasionally. Four out of 6 of these patients reported effectiveness of PPI in preoperative phase. 1 patient showed no improvement of the GERD symptomatology while the remaining 1 patient did not suffer from of any symptoms during preoperative phase, but the patient experienced de novo GERD symptomatology (3.12 %).”
- The presentation of the results regarding PPI medication is confusing. Figure 2 should be accompanied by a table summarizing the PPI medication pre- and post-op (frequent, occasional user and de novo cases after BariClip.
Thank you very much for your remark. According to your recommendation the following additional table to summarize the results concerning the GERD evolution was included in the revised form of the manuscript:
|
|
Preoperatively |
One year postoperatively |
|
GERD with systematic PPI use |
8 patients |
1 patient |
|
GERD with occasional PPI use |
4 patients |
6 patients |
|
De novo GERD |
- |
1 patient |
Minor points
- Please define GERD abbreviation in the title
Thank you for the comment, the title was modified. Introducing the abbreviation would become too long.
- lines 14-15 (abstract) please clarify the meaning
We apologies but we don’t understand the comment. The entire manuscript was reviewed thoroughly and the abstract seems to us very clear.
- line 45 gastric banding or gastric bypass?
One of the references is related to the gastric bypass and its long term complications and the other to the gastric band and its quality of life.
- The introduction should include more references and give a better presentation of Gerd associated with bariatric surgery.
According to your suggestion, the following references were included in the revised form of the manuscript:
Assalia A, Gagner M, Nedelcu M, Ramos AC, Nocca D. Gastroesophageal Reflux and Laparoscopic Sleeve Gastrectomy: Results of the First International Consensus Conference. Obes Surg. 2020 Oct;30(10):3695-3705.
Bou Daher H, Sharara AI. Gastroesophageal reflux disease, obesity and laparoscopic sleeve gastrectomy: The burning questions. World J Gastroenterol. 2019 Sep 7;25(33):4805-4813
Guzman-Pruneda FA, Brethauer SA. Gastroesophageal Reflux After Sleeve Gastrectomy. J Gastrointest Surg. 2021 Feb;25(2):542-550
- The list of complication can be presented in a table.
We agree with your comment but to include an additional table we need to have more columns. In the actual form the complications are better explained in the current text.
- Reference should be corrected (for example refs 7 and 8)
The entire list of references was revised and completed with new references in the revised form
- The abstract should be edited for clarity.
The entire manuscript has had several revision and we hope the new revised form is more clear for you and the readers.
Round 2
Reviewer 1 Report
The manusctipt is well revised according to the reviewers' suggestions .
Author Response
We thank the reviewer for the positive feedback. We are convinced that by the modifications done to the manuscript according to your previous suggestions we have highly improved the quality of our paper.
Reviewer 2 Report
The authors addressed most of my concerns.
I agree that the limited size of the cohort precludes from statistical analysis and leads to the descriptive presentation of the results. However, this should be discussed in the manuscript both in the methods section (as discussed in the authors' reply) and in the discussion section as it represents a major limitation of the manuscript.
Moreover, please check the references in the text (references 4-6 are misplaced).
Author Response
The authors addressed most of my concerns.
We thank the reviewer for your fair and very constructive feedback. We have done the appropriate modifications according to our experience and convictions. We are convinced that by the modifications done brought to the manuscript according to your suggestions we have highly improved the quality of our paper.
I agree that the limited size of the cohort precludes from statistical analysis and leads to the descriptive presentation of the results. However, this should be discussed in the manuscript both in the methods section (as discussed in the authors' reply) and in the discussion section as it represents a major limitation of the manuscript.
Thank you for the remark. According to your suggestion and our previous discussions, the following paragraph was added to the revised form of the manuscript:
“Another limitation of the current study is represented by the reduced number of patients included, as for every new innovative bariatric procedure. Thus, we decided to use only descriptive analytics for the 12 patients with reflux symptoms and any statistical correlation was considered futile.”
Moreover, please check the references in the text (references 4-6 are misplaced).
Thank you very much for your remark and we apologies for this error. These references were corrected, and the entire list was reviewed accordingly.